# Impact of the Hydroelectric Dam on Aquifer Recharge Processes in the Krško Field and the Vrbina Area: Evidence from Hydrogen and Oxygen Isotopes

**Branka Trček *** and **Beno Mesarec**

Faculty of Civil Engineering, Transportation Engineering and Architecture, University of Maribor, SI 2000 Maribor, Slovenia
* Correspondence: branka.trcek@um.si

**Abstract:** The impact of the damming of the Sava river for the Brežice hydroelectric power plant on the rise of the groundwater level was studied in the intergranular aquifer of the Krško field and the Vrbina area, Slovenia. The study is based on the application of hydrogen and oxygen isotopes (18O, 2H and 3H). Parameters were determined for precipitation, surface water, and groundwater for periods before and after the filling of the accumulation basin, with the aim of evaluating the groundwater–surface water interaction and to elucidate the impact of the hydroelectric dam on aquifer recharge processes. The results show the proportions of the surface water component in groundwater sampled from four wells at high and low water conditions, separately for the period before and after the filling the accumulation basin. After filling the accumulation basin, the proportion of the Sava river component at high water conditions increased from 60% to 80% in the Brege and Drnovo wells (drinking water resources), from 50% to 80% in the Cerklje well and to almost 100% in the near-river NEK well. Combined with previous studies, the results provide important information about the direction of groundwater flow in the aquifer and improve the conceptual model of the study site.

**Keywords:** stable isotopes; tritium; dam; surface water; groundwater

## 1. Introduction

In accordance with Slovenia's strategic goals for sustainable energy use, the construction of hydroelectric power plants (HPP) on the Sava river was planned. The HPP Brežice was built upstream of the city of Brežice (Figure 1). It was put into operation at the end of 2017. Consequently, the damming of the Sava river had an impact on the rise of the groundwater level in the intergranular aquifer of the Krško field and the Vrbina area, which provides the inhabitants of the region with water essential for their drinking and sanitary needs. The above findings prompted a detailed hydrological and hydrogeological investigation to characterize the groundwater body. Understanding the relationship between groundwater recharge and discharge is one of the most important aspects of protecting groundwater resources. Ground and surface water are fundamentally interconnected, and it is often difficult to separate the two. As they recharge each other, they can also contaminate each other, which should be considered in management planning, especially when water is supplied for drinking.

Besides direct methods, indirect methods were applied to study the groundwater flow and solute transport in the Krško field and the Vrbina area. A mathematical model of the study site was created to simulate the recharge processes of the aquifer depending on the water levels in the dammed area and in the drainage channels [1–3]. In addition, the impact of the hydroelectric dam on aquifer recharge processes in the Krško field and the Vrbina area was studied using hydrogen and oxygen isotopes, which is described in this manuscript.

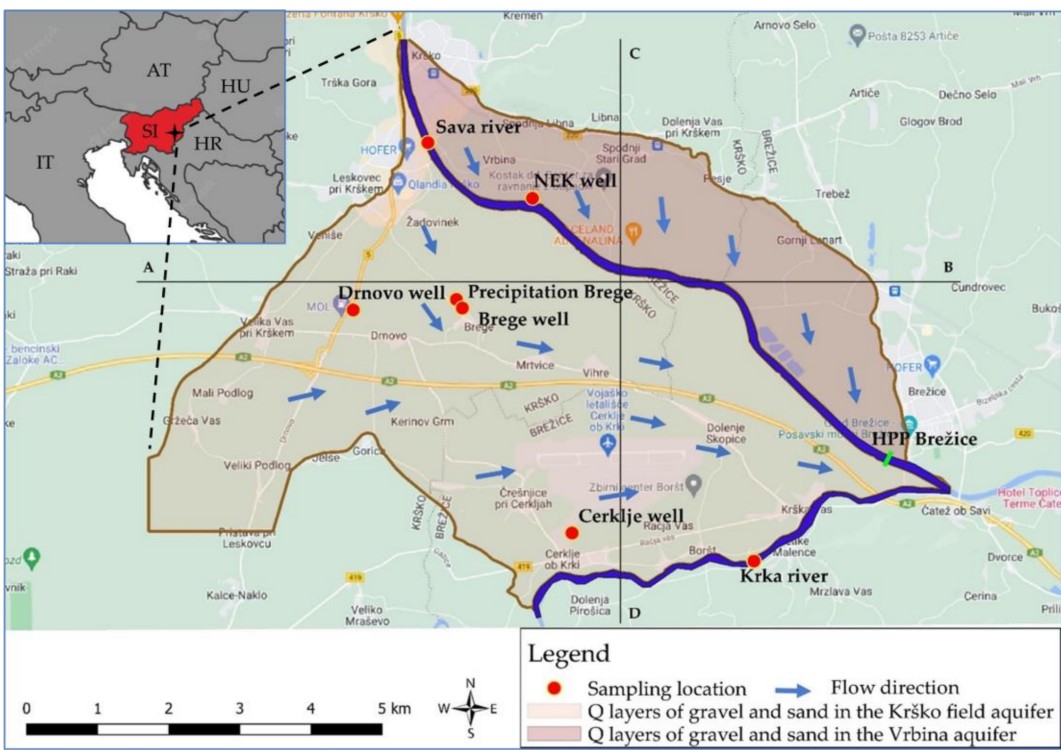

**Figure 1.** Field site with a simplified geological composition of the main groundwater flow directions (modified after [3]) and locations of water samples in relation to water type.

Stable ($^{16}$O, $^{17}$O, $^{18}$O, $^{1}$H, $^{2}$H) and radioactive ($^{3}$H) isotopes in water molecules are powerful tools for the tracing the path of water molecules in the water cycle. The isotopic composition of groundwater varies at a given sampling location in the aquifer as a function of precipitation, surface water and groundwater inflow [4–9]. Stable isotopes of oxygen and hydrogen ($^{18}$O and $^{2}$H) are particularly useful for monitoring the transport of infiltrated precipitation and surface water in the aquifer because they usually do not change after entering the low-temperature aquifer [8–14]. After water enters the aquifer, only physical processes such as diffusion, dispersion, mixing, and evaporation alter the groundwater's isotopic composition. Therefore, $\delta^{18}$O and $\delta^{2}$H of water in the low-temperature groundwater system of the Krško field and the Vrbina area can be considered conservative.

Since $^{3}$H has a half-life of 12.3 years, it is a very suitable tracer to determine the age of groundwater that has been in the aquifer for less than 50 years [11,15–17]. Radioisotopic decay provides information on the circulation time of water and thus on the renewability of groundwater [18,19]. However, occasionally extreme values of $^{3}$H activity concentration are caused by episodic emissions of technogenic $^{3}$H on the European continent [20]. Liquid and atmospheric discharges from the Krško nuclear power plant (NEK) are an additional anthropogenic source of $^{3}$H in the study site [21,22] and represent an important tracer for the study of groundwater flow and solute transport in the Krško field and Vrbina aquifer.

Studies of the isotopic composition of $^{18}$O and $^{2}$H, and $^{3}$H activity concentration in waters have wide application in characterization of water bodies. As reviewed by Bronić and Barešić [4], and Vreča and Kern [5], they allow the determination of the mean residence time of water in aquifers, surface water–groundwater interactions, recharge areas, connections between aquifers, origin of groundwater, and dynamics of processes in surface waters. Several studies have been carried out during the last decade where isotope composition of precipitation, surface water, and groundwater were applied for improvement of the conceptual model of porous aquifer systems (e.g., [23–30]). This refers to intergranular aquifers hydraulically connected to the river, as is the case in the Krško field and Vrbina aquifer, and points to the application of the research methodology for the

study of surface water–groundwater interactions as a result of the damming of the Sava river for HPP Brežice. Some manuscripts present studies on the impact of river dams on surface and groundwater bodies (e.g., [31–38]). These studies mostly focus on large river systems after damming the river under different climatic conditions. Their results describe the effects of damming on the river water cycle, ecosystem, and biogeochemical processes, and are relevant for future studies in the catchment area, as our study investigates the hydrodynamic conditions in the aquifer one hydrological year before and after the filling of the accumulation basin for the HPP Brežice. The first aim of our investigations was to identify and interpret the $^{18}$O and $^{2}$H composition and the $^{3}$H activity concentration in precipitation, surface water, and groundwater in the catchment area of the Krško field and Vrbina aquifer for the periods before and after the filling of the accumulation basin. The evaluation of ground water–surface water interaction was put into focus. The second aim was to determine the shifts of the studied parameters in the mentioned monitoring periods and to elucidate the impact of the hydroelectric dam on aquifer recharge processes. The final aim was to integrate the isotopic results with the previous mathematical model results and discuss them in terms of different flow paths and possible sources of contamination.

## 2. Materials and Methods

### 2.1. Study Site

The Krško field and the Vrbina area is a part of the south-eastern region of Slovenia. The study site is located in the Krško basin between the Sava and Krka rivers (Krka is a tributary of Sava; Figure 1). The relief is flat with altitudes between 140 and 160 m. In the north and in the south, it is bordered by the hills Bizeljsko and Gorjanci. The region has a moderate continental climate with strong sub-Pannonian characteristics [39]. The average annual precipitation amount is 1100 mm and the average winter and summer temperatures are −1 and 21 °C, respectively.

The basin is filled with Quaternary alluvial deposits from the Sava and Krka rivers and local streams from the surrounding hills [1–3]. The rivers deposited mainly coarse-grained gravels and sands, while the local tributaries deposited fine-grained sands, silts, and clays. These sediments form an unconfined intergranular aquifer used for public water supply, industry, and agriculture. The alluvial aquifer receives a natural recharge mainly from precipitation infiltrating into the catchment and the Sava river. However, some groundwater is also discharged from the aquifer into the river. The Sava river recharges the Krško field aquifer even at low water conditions in sections between Krško, Žadovinek and Brege, and between Vihre and Skopice, while the aquifer between Skopice and the confluence of the Sava and Krka rivers drains into the Sava ([1]; Figure 1). The same applies to the Vrbina aquifer. In the upper section, the river recharges the aquifer, while in the lower section the aquifer drains into the river. On the other hand, the Krško field aquifer between Boršt and the confluence of the Sava and Krka rivers also drains into the Krka river.

The recharge–discharge processes of the aquifer are reflected in the main directions of groundwater flow (Figure 1). In general, groundwater flows in the NW–SE direction, parallel to the Sava river. Local changes occur in the drainage areas and near pumping stations and dams. The groundwater table in the study site oscillates 3–7 m below the surface. The depth of the saturated zone at high water conditions of the Krško field and the Vrbina aquifer is 4.5–12 m and 2–8 m, respectively. The aquifer permeability is good and ranges from $1.2 \times 10^{-4}$ to $7 \times 10^{-2}$ m/s, while the effective porosity is estimated at 8–21% [1–3].

The Neogene intergranular aquifer lies beneath the alluvial deposits of the Sava and Krka rivers and their tributaries [1–3]. It consists of very-low-permeability to practically impermeable Pliocene sands and clays and Miocene marls. Their depth exceeds 100 m. The Neogene aquifer has up to 100 times lower permeability than overlying Quaternary aquifer. Mesozoic sedimentary rocks, mainly Triassic dolomites, are in its basement and form the third aquifer, the so-called channel-fractured aquifer. The first and second aquifers are partially hydraulically connected, while there is only an indirect connection with the

third aquifer. Figure 2 shows a detailed cross-section through the Quaternary and Neogene hydrogeological structure of the Krško field and Vrbina aquifer.

**Figure 2.** Cross sections (**A**,**B**) and (**C**,**D**) of the hydrogeological structure of the Krško field and Vrbina aquifer (modified after [3]; further information can be found in Figure 1).

*2.2. Sampling*

Isotope monitoring was performed one hydrological year before the damming of the Sava river for the HPP Brežice and one hydrological year after the filling of the accumulation basin, after the groundwater level had stabilized. Monitoring included 24 phases of sampling, 12 before and 12 after filling the accumulation basin. Sampling was performed at monthly intervals under varying water conditions. A total of eight sampling locations were included in the monitoring. They were selected according to the water type: precipitation, surface water, and groundwater. (Figure 1, Table 1).

According to the instructions of the International Atomic Energy Agency, two buried totalizers were built for precipitation sampling [40]. The first was placed next to the Brege drinking water pumping station at an elevation of 154 m above sea level. The second was placed next to the precipitation station of the Slovenian environmental agency in Puste Ložice at an elevation of 564 m above sea level. The precipitation collected during the sampling phases was sampled for the analysis of $\delta^{18}O$ and $\delta^{2}H$ as well as for the $^{3}H$ activity concentration. The available isotopic data include more than 70% of the total precipitation amount collected per year, which is the IAEA requirement [41].

Surface water and groundwater were sampled for the same analysis. Surface water samples were collected from two locations, the Sava and Krka rivers (Figure 1, Table 1). Four sampling locations represent groundwater, two of which are drinking water resources—the Brege well and the Drnovo well. Samples were taken during continuous production of the

wells from built-in taps, after pH, temperature, and electrical conductance had stabilized while pumping.

**Table 1.** Locations, types, and names of water samples in the study site (further information can be found in Figure 1).

| Name of Sampling Location | Well Depth (m) | Water Type | Coordinates | |
| --- | --- | --- | --- | --- |
| | | | x | y |
| Sava river | | Surface water | 538,269.50 | 90,003.31 |
| Krka river | | | 540,615.96 | 82,077.78 |
| NEK well | 10.0 | Groundwater | 540,489.49 | 88,349.86 |
| Cerklje well | 8.8 | | 540,938.88 | 83,055.74 |
| Brege well | 13.1 | | 539,329.55 | 86,882.22 |
| Drnovo well | 18.8 | | 537,823.41 | 86,333.60 |
| Precipitation Brege | | Precipitation | 539,316.85 | 86,901.27 |
| Precipitation Puste Ložice | | | 534,128.64 | 100,937.54 |

A total of 192 water samples were collected. All samples were stored in HDPE plastic bottles without being treated. Samples were transported to the laboratory of Joanneum Research–AquaConSol GmbH, Graz, Austria within 24 h after sampling.

Isotopic analyses of stable isotopes of $^{18}O$ and $^{2}H$ were made in the laboratory of Joanneum Research–AquaConSol GmbH [42–44]. Measurements of the stable isotopic composition of $^{18}O$ and $^{2}H$ are conventionally reported in terms of a relative value δ (‰):

$$\delta x = (R_x / R_{st} - 1) \cdot 1000 \tag{1}$$

where $R_x$ is the isotope ratio (e.g., $^{2}H/^{1}H$ and $^{18}O/^{16}O$) in the substance x and $R_{st}$ is the isotope ratio in the corresponding international standard substance [45]. Stable oxygen and hydrogen isotopic ratios are reported relative to the VSMOW2 (Vienna Standard Mean Ocean Water 2) and SLAP2 (Standard Light Antarctic Precipitation 2) standards with an overall precision of ±0.1 and ±1‰, respectively. Negative or positive delta values mean that the isotope ratio of the sample is lower or higher relative to a standard [45,46].

The measurements of $^{3}H$ activity concentration were performed in the isotope laboratory HYDROSYS–Water and Environmental Protection Developing Ltd., Budapest, Hungary. The values are expressed in absolute concentrations as tritium units (TU), where one TU corresponds to one $^{3}H$ atom per 1018 atoms of hydrogen $^{1}H$ or an activity of 0.118 Bq/kg in water.

*2.3. Data Analysis*

Statistical processing of data was performed with Statistica 7 (StatSoft Inc., Tulsa, OK, USA). Tukey's boxplots were applied for the graphical data analysis [47]. The whiskers extend only to the last observation within one step beyond either end of the box. A step equals 1.5 times the interquartile range. Observations between one and two steps from the box in either direction are plotted individually as outside values (occurring less than once in 100 times for data from a normal distribution) and as far-out values (occurring less than once in 300,000 times for a normal distribution) in the case when they are farther than two steps beyond the box.

It is well known that $\delta^{18}O$ and $\delta^{2}H$ are linearly dependent. Their relationship was studied based on the global meteor line (GMWL), which was first described by Craig [48] and later improved by Rozanski et al. [49]:

$$\delta^{2}H = 8.13 \cdot \delta^{18}O + 10.8 \tag{2}$$

The GMWL explains the relationship between $\delta^{18}O$ and $\delta^{2}H$ in precipitation on a global scale, but its slope and/or intercept may change locally owing to differences in

climate and geographic characteristics [49–51]. Therefore, the use of local meteorological water lines (LMWL) is more appropriate for water body characterization, considering that precipitation is an input to the aquifer and knowledge of $\delta^{18}O$ and $\delta^2H$ of precipitation is a prerequisite for groundwater studies. A slope of a LMWL greater than eight indicates multiple moisture circulation, while a value less than eight indicates greater moisture loss through evaporation.

There are different ways to calculate the LMWL [30,51]. Because of the small number of samples, ordinary linear regression analysis based on the method of seasonally weighted mean values was used in our studies [49]. The isotopic composition of water is weighted according to the formula for the example of the isotopic composition of $^{18}O$ [26]:

$$\delta^{18}O = (\Sigma_{i=1}^n P_i\delta_i^{18}O)/(\Sigma_{i=1}^n P_i) \tag{3}$$

where:

$\delta^{18}O$—weighted average of the oxygen isotopic composition of precipitation/sampled water;
$P_i$—precipitation amount/water volume of the sample (i);
$\delta_i^{18}O$—oxygen isotopic composition of the sample (i).

The LMWL of the study site was compared with the $\delta^{18}O$ and $\delta^2H$ of surface water and groundwater together with the GMWL and the LMWLs of the nearest cities with long-term data—Ljubljana, $\delta^2H = 7.94\delta^{18}O + 9.76$ [51]; Zagreb, $\delta^2H = 7.68\delta^{18}O + 6.2$ [30]; Central Italy, $\delta^2H = 7.0\delta^{18}O + 5.6$ [52].

The intercept of the GMWL is also called deuterium excess or d-excess, and is defined as [53]:

$$d\ (\text{‰}) = \delta^2H - 8.13 \cdot \delta^{18}O \tag{4}$$

The parameter is an indicator of climate sensitivity at the source of humidity, as well as along the trajectory of air masses into the atmosphere [54]. It can be depicted visually as an index of deviation from the GMWL. Vapor generated over a closed basin with restricted communication, such as that of the Mediterranean sea, is characterized by a higher d-excess value (about 20‰), compared with Atlantic air masses with a lower d-excess value (about 10‰) [55,56].

Assuming that the studied aquifer is recharged by the Sava river and by precipitation, the two-component model was used to determine their proportions. The proportions of surface water and precipitation components in the sampled groundwater were estimated based on two mass balance equations, similar to those used in the study of the unconfined intergranular aquifer of Zagreb [18]:

$$f_{sw} + f_p = 1 \tag{5}$$

$$f_{sw} \times \delta^{18}O_{sw} + f_p \times \delta^{18}O_p = \delta^{18}O \tag{6}$$

where $f_{sw}$ and $f_p$ are the fractions of surface water and precipitation, respectively, and $\delta^{18}O_{sw}$ and $\delta^{18}O_p$ are the isotopic compositions of $^{18}O$ in surface water and precipitation, respectively.

## 3. Results

### 3.1. Isotopic Composition of $^{18}O$ and $^2H$

The mean, maximum, and minimum values of isotopic composition of $^{18}O$ and $^2H$, as well as the d-excess in the sampled waters, are given in Table 2 for the periods before and after the filling of the accumulation basin for HPP Brežice. The mean values of $\delta^{18}O$ and $\delta^2H$ in precipitation varied from −9.05‰ to −6.26‰ and from −59.65‰ to −37.21‰, respectively. The mean values of $\delta^{18}O$ and $\delta^2H$ in surface waters ranged from −9.62‰ to −8.98‰ and from −63.94‰ to −58.89‰, respectively. In groundwater, the parameters varied from −9.34‰ to −8.90‰ and from −63.93‰ to −58.56‰, respectively. These values are comparable to the results obtained by Mezga et al. [24] for the study site.

**Table 2.** Median, minimum, and maximum values of oxygen and hydrogen stable isotopic composition, d-excess, and tritium activity concentrations before and after the filling of the accumulation basin for HPP Brežice.

| | Sample Name | Median before/after Filling | Minimum before/after Filling | Maximum before/after Filling |
|---|---|---|---|---|
| $\delta^{18}O$ (‰) | Krka river | −9.62/−9.22 | −10.38/−10.08 | −9.09/−8.78 |
| | Sava river | −9.18/−8.98 | −9.65/−9.63 | −8.83/−8.54 |
| | NEK well | −9.24/−8.95 | −9.48/−9.48 | −8.83/−8.57 |
| | Cerklje well | −8.94/−8.90 | −9.19/−9.08 | −8.78/−8.44 |
| | Drnovo well | −9.34/−9.14 | −9.52/−9.19 | −9.14/−8.83 |
| | Brege well | −9.27/−9.01 | −9.50/−9.18 | −9.09/−8.84 |
| | Precipitation Brege | −8.00/−6.26 | −9.99/−12.35 | −6.43/−5.09 |
| | Precipitation Ložice | −9.05/−6.84 | −11.78/−13.56 | −6.00/−5.72 |
| $\delta^{2}H$ (‰) | Krka river | −63.9/−61.1 | −70.6/−68.2 | −60.4/−56.4 |
| | Sava river | −60.1/−58.9 | −64.2/−64.6 | −56.3/−54.3 |
| | NEK well | −61.2/−58.6 | −63.1/−64.0 | −59.4/−54.8 |
| | Cerklje well | −60.1/−59.7 | −62.7/−61.0 | 61.00/−58.0 |
| | Drnovo well | −63.9/−60.7 | −66.2/−61.2 | −62.2/−59.8 |
| | Brege well | −63.1/−61.1 | −65.2/−61.6 | −62.3/−59.7 |
| | Precipitation Brege | −54.8/−37.2 | −75.9/−84.9 | −24.2/−31.1 |
| | Precipitation Ložice | −59.6/−41.5 | −77.6/−96.0 | −37.6/−31.0 |
| d-excess (‰) | Krka river | 12.5/12.8 | 12.0/11.6 | 13.9/13.8 |
| | Sava river | 13.0/13.0 | 10.5/11.9 | 13.9/14.0 |
| | NEK well | 12.7/13.0 | 10.9/11.9 | 13.7/13.7 |
| | Cerklje well | 11.5/11.4 | 9.2/9.5 | 12.6/11.9 |
| | Drnovo well | 11.0/13.2 | 9.9/10.9 | 11.7/15.1 |
| | Brege well | 10.8/11.3 | 9.2/10.3 | 12.5/12.0 |
| | Precipitation Brege | 9.9/10.7 | 6.8/8.9 | 15.5/15.4 |
| | Precipitation Ložice | 11.0/12.4 | 5.4/9.1 | 16.6/14.7 |
| $^{3}H$ activity concentration (TU) | Krka river | 5.9/5.0 | 5.4/3.6 | 6.7/7.1 |
| | Sava river | 5.1/5.9 | 3.9/3.8 | 6.3/24.3 |
| | NEK well | 5.9/6.0 | 5.0/3.9 | 8.9/14.8 |
| | Cerklje well | 6.8/5.8 | 5.3/2.2 | 8.2/8.0 |
| | Drnovo well | 13.8/8.3 | 9.7/4.2 | 15.6/14.0 |
| | Brege well | 15.7/14.3 | 13.4/4.1 | 18.0/17.9 |
| | Precipitation Brege | 16.7/15.7 | 10.2/4.7 | 22.7/22.7 |
| | Precipitation Ložice | 8.1/5.9 | 4.3/1.7 | 12.6/11.8 |

The mean calculated value of d-excess in precipitation, surface water, and groundwater varied from 9.95‰ to 12.44‰, from 12.54‰ to 12.99‰, and from 10.77‰ to 13.04‰, respectively (Table 2). Parlov et al. [18] reported similar values of the parameters for observation wells in Zagreb. Values of d-excess around 10‰ are typical for continental meteoric water [48] and can be attributed to precipitation of Atlantic origin [57]. This precipitation predominates in the Krško and Zagreb area, as precipitation from the Mediterranean region is characterized by higher d-excess values of up to 25‰ [49].

The oscillation of $\delta^{18}O$ in water samples in the catchment area of the Krško field and Vrbina aquifer before and after the filling of the accumulation basin for the HPP Brežice is presented in Figure 3, together with the daily precipitation amounts in Brege and Puste Ložice. In the period before the accumulation basin for the HPP Brežice was filled, 1100 mm of precipitation fell in Pusti Ložice and 941 mm in Cerklje/Brege. At both precipitation stations, the share of precipitation in spring, summer, and autumn was similar (approximately 30%), while the share of winter precipitation was only 10%. In the period after the accumulation basin for the HPP Brežice was filled, 1174 mm of precipitation fell in Pusti Ložice and 995 mm in Cerklje/Brege, which is 6% more than in the first period of isotope monitoring. At both precipitation stations, the share of winter precipitation was the smallest (approximately 11%), the share of summer precipitation was the largest

(approximately 45%), and a similar amount of precipitation fell in spring and autumn (approximately 22%). The meteorological data described allow isotopic data to be compared between two observation periods and provide information about hydrological conditions in the aquifer, particularly the differences between dry-season and high-water conditions.

The isotopic composition of $^{18}$O in precipitation depends on temperature changes with latitude, variation in the amount of moisture from which precipitation forms, the area of origin of the moisture, and its atmospheric evolution [8,9,27]. Higher $\delta^{18}$O values are associated with higher temperature and vice versa. The amplitude of the $\delta^{18}$O oscillation is highest in precipitation; it then attenuates in surface and groundwater with the residence time of water. It is the smallest in groundwater with the longest retention time. In both monitoring periods, the mean values and ranges of $\delta^{18}$O of surface water are lower than precipitation (Table 2, Figure 3), which enables the study of the mixing of precipitation and river water in the subsoil.

$\delta^{18}$O oscillated similarly in groundwater of the Brege and Drnovo wells in both periods, before and after the filling of the accumulation basin for the HPP Brežice (Figure 3), indicating similar components of groundwater flow. The parameter also varied in a similar way in groundwater of the NEK well and in the Sava river, especially in the period after the accumulation basin was filled, which reflects the predominant river component in groundwater of the NEK well. The differences were more evident only in the dry season.

Before filling the accumulation basin, groundwater of the Cerklje well and the Krka river were characterized by the highest and lowest $\delta^{18}$O values, respectively (Figure 3). The same applies to the Krka river in the period after the accumulation basin was filled, reflecting the separate hydraulic system. On the other hand, the oscillation trend of $\delta^{18}$O in groundwater of the Cerklje well approached the trend of the Sava river, illustrating the increased recharge of surface water into the aquifer.

The linear relationship between $\delta^{18}$O and $\delta^2$H in precipitation is presented in Figure 4 by the LMWL-Krško: $\delta^2$H = 7.58 $\delta^{18}$O + 7.82 ($R^2$ = 0.98). The LMWL-Krško differs from the GMWL both in slope and intercept, which depends on climatic characteristics, i.e., evaporation and condensation processes [8,19,30,51]. Due to the small number of samples, the LMWL-Krško in Figure 4 is compared with the LMWLs of the nearest cities with long-term data: Ljubljana [51], Zagreb [30], and northern Italy [52]. The LMWL-Krško best matches the LMWL-Northern Italy, reflecting the similarity of climatic characteristics.

The LMWL of the study site is compared to the $\delta^{18}$O and $\delta^2$H in surface water and groundwater in Figure 5a,b, along with the GMWL and the LMWLs of Ljubljana [51], Zagreb [30] and central Italy [52].

All surface water samples are plotted above the LMWL-Krško, between the LMWL-Krško and the LMWL-central Italy. This shows that a larger part of the recharge area of the rivers Sava and Krka is located upstream of the study site and is under the influence of a different climate. Therefore, the influence of the recharge area in the study site is small. The $\delta^{18}$O and $\delta^2$H values of groundwater are distributed around the LMWL-Krško, which confirms their meteoric origin. All samples are plotted between GMWL and LMWL-Central Italy.

The average $\delta^{18}$O and $\delta^2$H of groundwater is generally equal to the weighted average $\delta^{18}$O and $\delta^2$H of waters that recharge it. Accordingly, Figure 6 shows the mean $\delta^{18}$O and $\delta^2$H of surface and groundwater and the weighted average precipitation values. The weighted averages of precipitation differ between the two periods of isotope monitoring—before and after the filling of the accumulation basin for HPP Brežice. The same is true for groundwater and surface waters. In the period after the filling of the accumulation basin, all sampled waters had higher mean values of $\delta^{18}$O and $\delta^2$H.

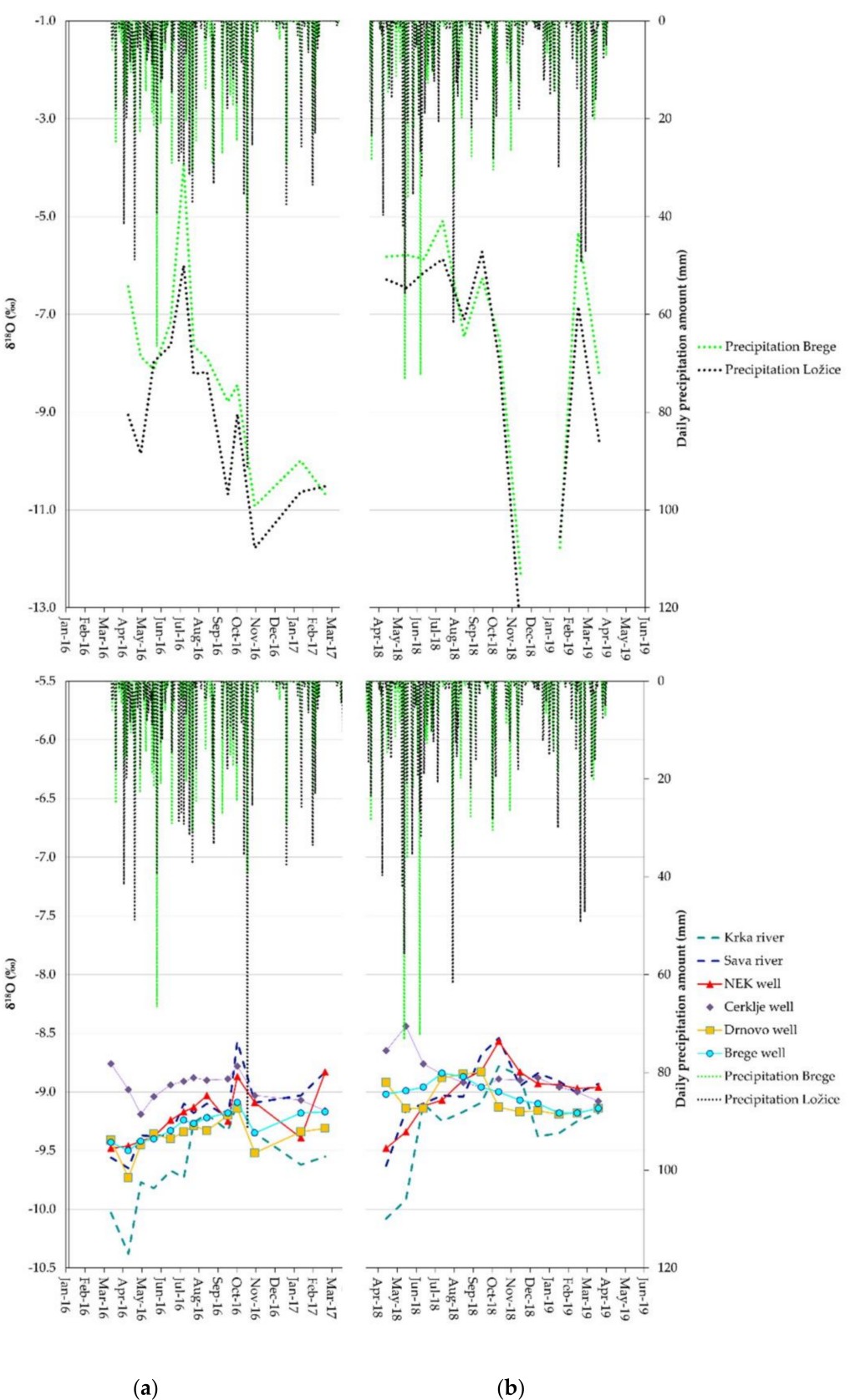

**Figure 3.** Oscillation of δ¹⁸O in water samples in the catchment area of the Krško field and Vrbina aquifer (**a**) before and (**b**) after the filling of the accumulation basin for the HPP Brežice in relation to the daily amount of precipitation in Brege and Puste Ložice.

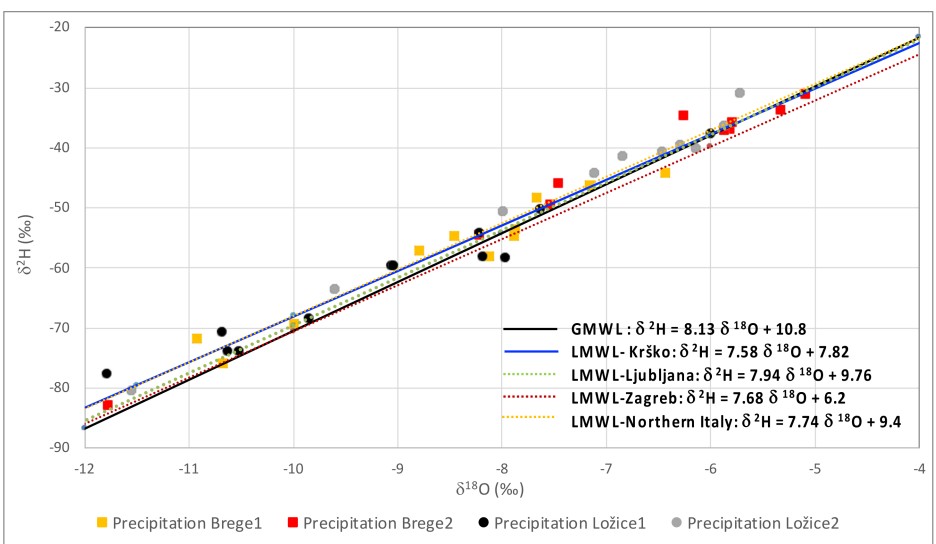

**Figure 4.** $\delta^{18}$O and $\delta^2$H in precipitation before (number 1 next to sample name) and after (number 2 next to sample name) the filling of the accumulation basin for the HPP Brežice.

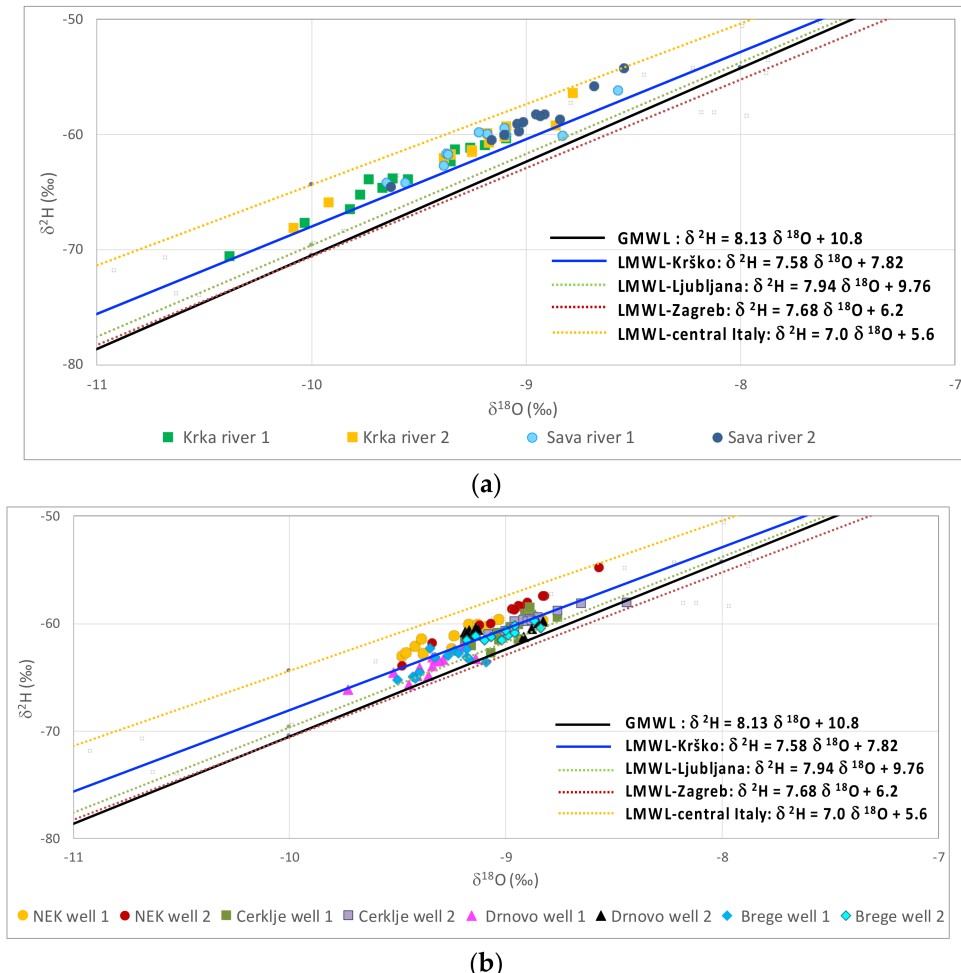

**Figure 5.** $\delta^{18}$O and $\delta^2$H in surface water (**a**) and groundwater (**b**) before (number 1 next to sample name) and after (number 2 next to sample name) the filling of the accumulation basin for the HPP Brežice.

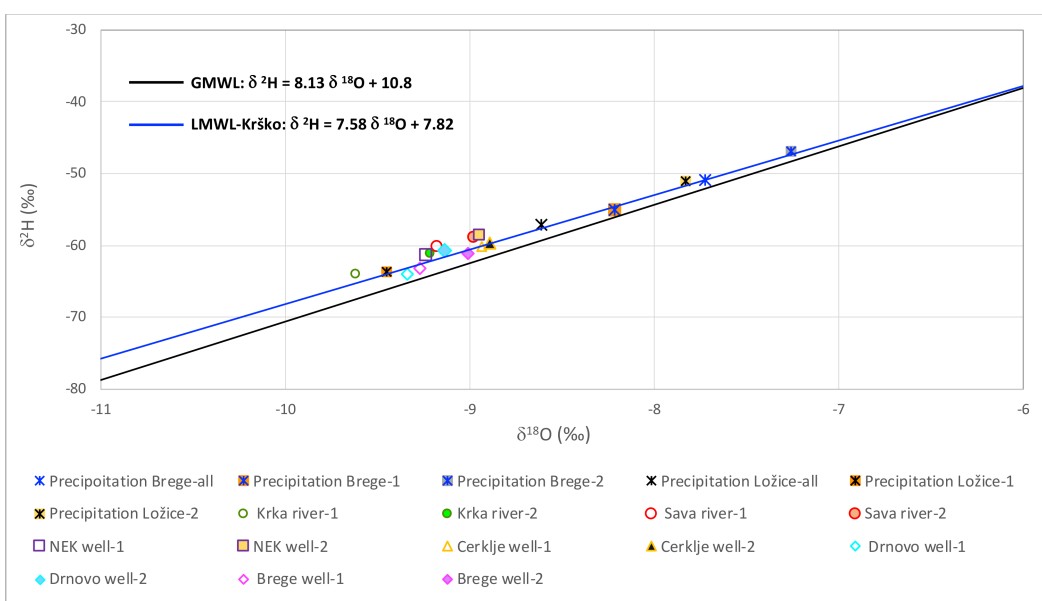

**Figure 6.** Median values of $\delta^{18}$O and $\delta^2$H in surface water and groundwater, and weighted averages of $\delta^{18}$O and $\delta^2$H in precipitation before (number 1 next to sample name) and after (number 2 next to sample name) the filling of the accumulation basin for the HPP Brežice.

Groundwater of the NEK, Cerklje, Brege, and Drnovo wells reflect the mixing of the Sava water and precipitation components (Figure 6). The largest proportion of the Sava water component is in the NEK well, since the mean $\delta^{18}$O and $\delta^2$H is closest to the river value. This is especially true for the period after the accumulation basin was filled, when the values of the Sava river and the NEK well overlap. The relations between the values of the Sava river and the wells before and after filling of the accumulation basin show, on the one hand, that the proportion of the Sava water component in the Brege well could increase after the filling of the basin and, on the other hand, that it could decrease in the Drnovo well. Undoubtedly, the share of the Sava water component in the Cerklje well has increased significantly.

The proportions of the surface water component in groundwater samples were investigated in more detail using the $\delta^{18}$O data. The results are presented graphically with boxplots (Figure 7). The boxplots clearly show the characteristics of the measured parameter, namely the range and distribution of the data, and allow comparison and separation of the individual sample locations. They illustrate the mean (the middle line of the box), the dispersion (the height of the box), the asymmetry (the ratio of the heights of two boxes), and the presence of extreme values (values outside and far outside).

Additionally, boxplots were used to determine the altitude effect at the study site. Namely, it is commonly observed that precipitation gradually becomes depleted in $^{18}$O and $^2$H isotopes as altitude increases [53]. This phenomenon results primarily from the cooling of air masses as they ascend a mountain range, accompanied by the dissipation of excess moisture [58].

Boxplots in Figure 7 show the $\delta^{18}$O of sampled waters for the periods before and after the filling of the accumulation basin for the HPP Brežice separately. In both periods, the mean $\delta^{18}$O of precipitation in Puste Ložice is approximately 1‰ lower than in Brege. It follows that the $\delta^{18}$O of precipitation decreases by 0.24‰ per 100 m above sea level, which can be used to determine the catchment area of the Krško field and Vrbina aquifer. The result is consistent with data from previous studies. It is reported that the altitude effect on precipitation ranges from −0.2‰ to −0.3‰ $\delta^{18}$O per 100 m on Slovenian territory [24,57], while the European average is −0.21‰ $\delta^{18}$O per 100 m [59].

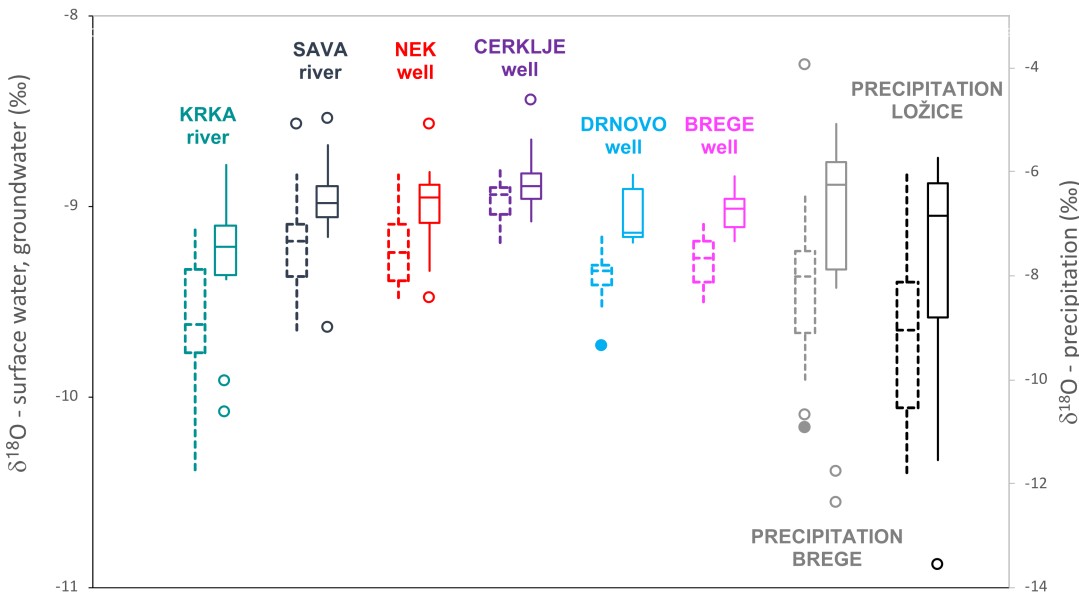

**Figure 7.** Distribution of $\delta^{18}$O with upper and lower quartile, median, and outside values in precipitation, surface water, and groundwater sampled before (dashed line) and after (solid line) the filling of the accumulation basin for the HPP Brežice.

Figure 7 indicates that all sampled waters had higher mean values of the $\delta^{18}$O after the accumulation basin for the HPP Brežice was filled, which can be explained by different precipitation inputs. A comparison of the boxplots of the sampling locations for the NEK well and the Sava river shows similar $\delta^{18}$O properties, which means that the Sava river water was a predominant component of groundwater exploited from the NEK well during both monitoring periods. The range of data was even more similar after the accumulation basin was filled, and the mean values differed only minimally (Table 2, Figure 7). It follows that the proportion of Sava water in groundwater near the NEK actually increased and was practically the only component of groundwater recharge, since the component of precipitation was negligible.

The boxplots of $\delta^{18}$O from the Brege and Drnovo wells (Figure 7), in connection with the data in Figure 3, reflect the similarity of groundwater in both periods, before and after the accumulation basin was filled. Thus, the two wells are recharged from same sources. In the first period, we can see in Figure 3 that the $\delta^{18}$O values of the discussed sampling locations were occasionally intertwined with the values of the Sava river, indicating that the river recharge the wells. In Figure 7, the data from the Brege and Drnovo sampling locations cover 60% of the data from the Sava sampling location. This means that the maximum proportion of Sava water component in groundwater of the Brege and Drnovo wells is about 60%. The distribution of $\delta^{18}$O means from the two wells shows that the average proportion of the surface water component is lower in groundwater. It is estimated that the average proportion of Sava water in groundwater of the Brege well before filling the accumulation basin was about 40%, while in groundwater of the Drnovo well, it was about 30%. In the period after the filling of the accumulation basin, the similarity of the $\delta^{18}$O of groundwaters of the Brege and Drnovo wells is even greater, especially at the end of the period (Figures 3 and 7). The data range of the two sampling locations does not differ, but their dispersion does, which reflects different recharge regime (Figure 7). The data from the Brege and Drnovo sampling locations cover 80% of the data from the Sava sampling location (Figure 7), which is 20% more than before the accumulation basin was filled. Thus, the maximum proportion of the Sava water component in the groundwater of the discussed wells after filling the accumulation basin is about 80%. Based on the comparison of the mean values of $\delta^{18}$O of the sampling locations, it is estimated that the average share of the Sava water component in groundwater of the Brege well remained unchanged at 40% after

filling the accumulation basin, while in groundwater of the Drnovo well, it decreased from 30% to 10%.

Before filling the accumulation basin, the highest $\delta^{18}O$ values were recorded in groundwater of the Cerklje well (Figures 3 and 7). It is estimated that the maximum proportion of the Sava water component in groundwater of the discussed well was 50% and the average share was 10%. The oscillation trend of the parameter in groundwater of the Cerklje well changed significantly after filling the accumulation basin (Figure 3). Based on the boxplots of the Cerklje well and the Sava river (Figure 7), it is estimated that the largest proportion of the Sava water component in the Cerklje well in the second monitoring period was 80%, while the average share was 25%.

*3.2. Tritium Activity Concentration*

The mean, maximum, and minimum values of $^3H$ activity concentration in the sampled waters are given in Table 2 for the periods before and after the filling of the accumulation basin for HPP Brežice.

The oscillation of $^3H$ activity concentrations in the sampled waters in the period before and after filling of the accumulation basin for the HPP Brežice is shown in Figure 8, together with the daily precipitation amounts in Brege and Puste Ložice, described in the beginning of Section 3.1. There was practically no precipitation in December 2016, so it is assumed that most of the water collected in the rain gauges is fog condensate.

$^3H$ activity concentrations in precipitation are much higher in Brege than in Puste Ložice and fluctuate strongly (Figure 8). The average values of the parameter before and after filling the accumulation basin were 16.7 and 15.7 TU in Brege, and 8.1 and 5.9 TU in Puste Ložice (Table 2). The maximum value of the parameter before filling the accumulation basin was 51.5 TU in Brege, while in the period after filling it was 22.7 TU. $^3H$ activity concentrations higher than 17 TU are attributed to the impact of discharges from the NEK [37,38]. The values in October 2016, which are 3.5 times higher than the values in precipitation of Puste Ložice, are striking.

The impact of $^3H$ discharges from the NEK on precipitation composition can also be traced in groundwater recharged by this precipitation (Figure 8). In the period before and after filling of the accumulation basin, they were detectable in groundwater of the Brege and Drnovo wells. In both sampling periods, occasional peaks of the parameter were observed in groundwater of the NEK well (Figure 8), dominated by the Sava water component (Figures 3 and 7). In the first monitoring period and at the beginning of the second period, they can be attributed to the effect of local precipitation. At the end of the second monitoring period, we note a difference in the trend of $^3H$ activity concentrations of the Sava river. Due to the damming of the river, its flow velocity has decreased, allowing a greater amount of local precipitation to be retained in it, which affects the increase in $^3H$ activity concentrations in the Sava River.

The characteristics of $^3H$ activity concentrations in the sampled waters are also presented in Figure 9 with boxplots. Groundwater in Brege and Drnovo shows the highest ranges and mean values of the parameter in both monitoring periods, indicating the impact of discharges from the NEK on the composition of precipitation recharging the mentioned groundwater sources.

The same applies to groundwater of the NEK well, except that the impact occurs occasionally, as shown by the outside and far-outside values. The greatest changes between the period before and after the filling of the accumulation basin are observed at the Sava river and the Cerklje well sampling locations. As mentioned in the previous paragraph, the retention time of the Sava river is longer after damming, so it is assumed that local precipitation influences the increase of $^3H$ activity concentrations in the river. It is also assumed that the $^3H$ activity concentration in groundwater of the Cerklje well has decreased due to the greater proportion of the Sava river component, as evidenced by the same mean values of the parameter at both sampling locations.

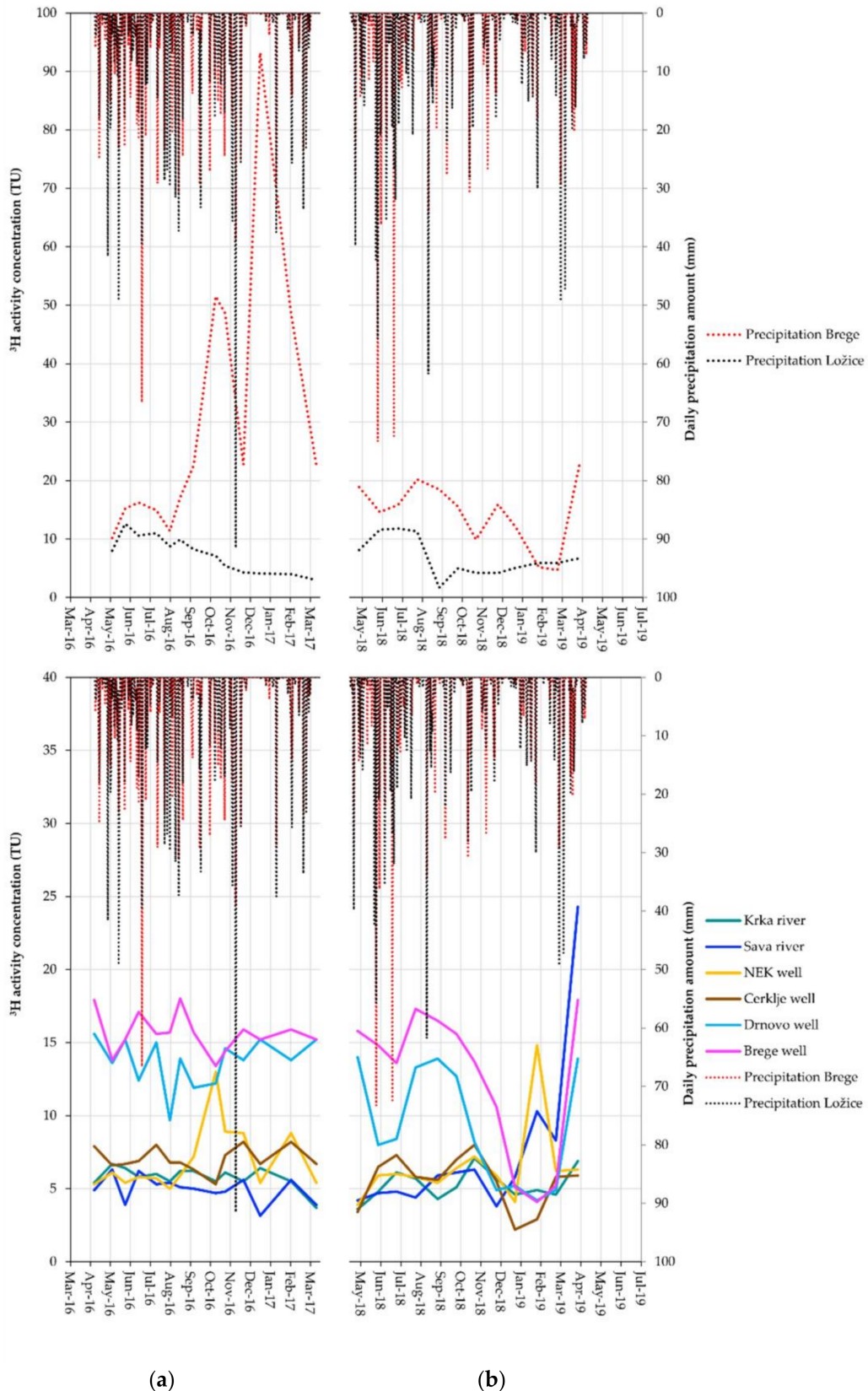

**Figure 8.** Oscillation of concentrations of tritium activity in water samples (**a**) before and (**b**) after the filling of the accumulation basin for the HPP Brežice in relation to the daily amount of precipitation in Brege and Puste Ložice.

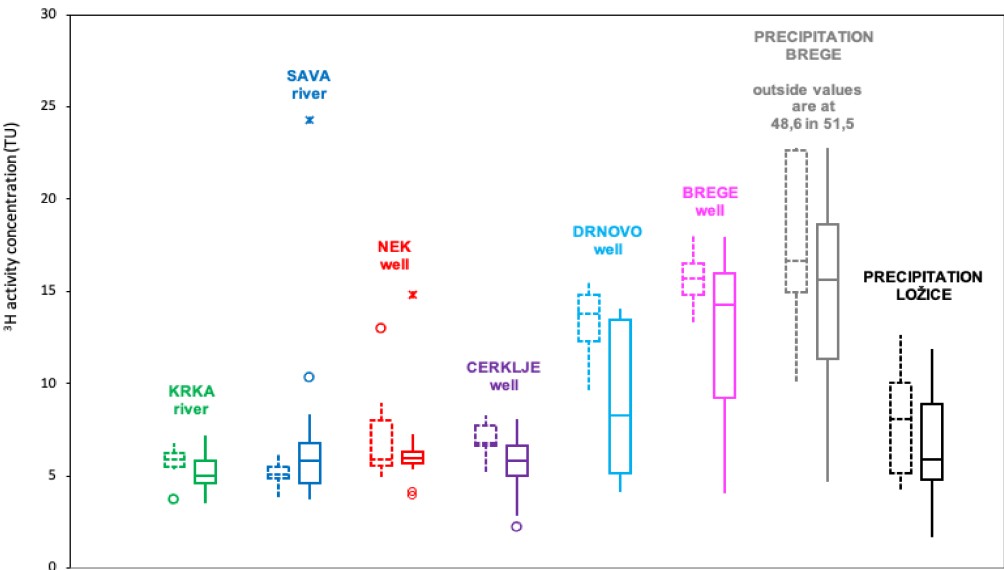

**Figure 9.** Distribution of concentrations of tritium activity with upper and lower quartile, median, outside, and far-out values in precipitation, surface water, and groundwater sampled before (dashed line) and after (solid line) filling of the accumulation basin for the HPP Brežice.

## 4. Discussion

The presented isotopic investigations serve as a tool to verify the simulations of the recharge processes in the Krško field and Vrbina aquifer after the filling of the accumulation basin for the HPP Brežice with a mathematical model designed in previous studies [1–3]. The simulations showed that the groundwater level in the Krško field aquifer would increase by 1 to 2.6 m during high water conditions after filling of the accumulation basin for the HPP Brežice. The largest differences between the measured high groundwater levels one year after the filling of the accumulation basin and the high groundwater levels from the simulations occurred in the upper half of the Krško field, near Drnovo and Cerklje [60,61], which can be explained by the results of our study.

After the accumulation basin was filled, the proportion of the Sava river increased at all sampling locations where groundwater was monitored during high water conditions. The largest changes in recharge processes after filling the accumulation basin were observed at the sampling locations Drnovo and Cerklje, at both high and low water conditions, which is reflected in the largest changes in the proportion of the Sava river component at the sampling locations.

Based on the data obtained in one hydrological year, it is estimated that the largest proportion of Sava water in groundwater of the Drnovo and Brege wells before the filling of the accumulation basin for the HPP Brežice was 60%, while the average proportion of river water in groundwater of the two wells was 30% and 40%, respectively. The maximum proportion of the Sava river in groundwater of the two wells was 80% after the accumulation basin was filled. The average proportion of Sava water in groundwater of the Brege well remained unchanged at 40%, while in groundwater of the Drnovo well it decreased from 30% to about 10%. The range of data from the two sampling locations was not different, but their dispersion was different, which highlights the different recharge regime. They follow the same recharge process at high water conditions and a different one at low/average water conditions, which can be explained by the groundwater flow direction in Figure 1. Unlike at high water conditions, the flow direction from Žadovinek to Drnovo is negligible at low water conditions, which was also confirmed with the mathematical model. The flow direction in the vicinity of Brege does not change, but the proportion of the river component increases by 40% at high water conditions, which is 20% more than before the filling of the accumulation basin.

The comparison of the studied parameters in the Krka river confirms that the surface water body is separated and does not recharge the aquifer, while the Sava River predominantly recharges the near-river NEK well. The proportion of the Sava river in the NEK well increased even more after the accumulation basin was filled, as evidenced by the matching of the mean values of the $\delta^{18}O$, $\delta^2H$, and $^3H$ activity concentrations of the two sampling locations. At the end of the second monitoring period, a difference in the trend of the $^3H$ activity concentration of Sava water was observed, which can be attributed to the effects of $^3H$ discharges from the NEK. Namely, the damming of the Sava led to a decrease in its flow velocity, which allowed greater amounts of local precipitation to be retained in the river and increased the above parameter. These changes were also reflected in groundwater of the Cerklje well, which shows that the proportion of Sava water at this location has increased. Based on the $\delta^{18}O$ and $\delta^2H$ data, it is estimated that the largest proportion of Sava water in groundwater of the Cerklje well before filling the accumulation basin was 50%, while the average proportion was 10%. It is estimated that the maximum proportion of Sava water in the well after the filling of the accumulation basin is 80%, while the average proportion is about 25%. The results show that the Sava river recharge increased in the NE direction from Brege to Cerklje (Figure 1) at low- and, especially, high-water conditions, while the Krka river represents an impermeable boundary. The presented findings were also confirmed by the mathematical model.

The integrated isotopic and simulation results provide important information on the direction of groundwater flow and solute/contaminant transport in the aquifer and improve the conceptual model of the study site, which enables appropriate planning of interventions in the Krško field and Vrbina aquifer and improves the strategy of water protection zones for drinking water resources.

## 5. Conclusions

The results of the isotope monitoring provide insight into the surface water–groundwater interactions in the Krško field and Vrbina aquifer for the period before and after the filling of the accumulation basin for the HPP Brežice. The comparison of the trends in $\delta^{18}O$, $\delta^2H$, and $^3H$ activity concentrations of the sampled water, as well as their range and dispersion, reflect the impact of the hydroelectric dam on aquifer recharge processes. After filling the accumulation basin, the proportion of the Sava river component increased to 80% in all observation wells at high water conditions, except in the near-river NEK well, where it was practically 100%. The largest changes in recharge processes were observed at the Drnovo and Cerklje wells, i.e., increases of 20% and 30%, respectively. At low water conditions, the average proportions of surface water components decreased by 20% in the Drnovo well and increased by 15% in Cerklje.

The combination of the isotopic data with the simulation results of the mathematical model provides information about the direction of groundwater flow and solute/contaminant transport in the study site under different hydrodynamic conditions. The estimates of the surface water–groundwater interactions after the filling of the accumulation basin for the HPP Brežice are based on isotope monitoring, which was performed only for one hydrological year; therefore, it is recommended to supplement the presented results with further research. Nevertheless, the presented results were essential for improving the conceptual model of the investigated water body, which is important for protection of water resources in the study area.

**Author Contributions:** Conceptualization, B.T.; methodology, B.T. and B.M.; validation, B.T. and B.M.; formal analysis, B.T. and B.M.; investigation, B.T. and B.M.; resources, B.T. and B.M.; data curation, B.M.; writing—original draft preparation, B.T.; writing—review and editing, B.T. and B.M.; visualization, B.T. and B.M.; supervision, B.T.; project administration, B.T.; funding acquisition, B.T. All authors have read and agreed to the published version of the manuscript.

**Funding:** This research was funded by INFRA Izvajanje Investicijske Dejavnosti d.o.o, grant number 13-ZJN/2015.

**Data Availability Statement:** Data sharing is not applicable to this manuscript.

**Acknowledgments:** The authors would like to thank GeoSi d.o.o. Geological Institute for management of the project and help in the field and with interpretation of the data.

**Conflicts of Interest:** The authors declare no conflict of interest.

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
