# Peer review of "Impact of the Hydroelectric Dam on Aquifer Recharge Processes in the Krško Field and the Vrbina Area: Evidence from Hydrogen and Oxygen Isotopes"

_water, doi:10.3390/w15030412_

Round 1

Reviewer 1 Report

Dear authors,

You manuscript deals with interesting subject but polishing is needed.

My remarks are:

Keywords: replace radioactive isotopes with tritium and delete aquifer

Abstract: rewrite Abstract in a way that you do not use same sentences from Introduction and Results

Why are you using through whole text this expression the evidence from hydrogen and oxygen isotopes? It is so non English style and hydrogeological…better is to say hydrogen-2 and oxygen-18 isotopes was used to indicate/determine/find out etc.

Line 75 delete the 18O and 2H composition leave just 18O and 2H and remove brackets

Too much basic information about usage of stable isotopes and tritium in hydrogeological research, delete unnecessary information. For example Tritium (3H) is a radioactive isotope. As a cosmogenic radionuclide it is formed in the 83 upper layer of the atmosphere during nuclear reactions of high-energy cosmic radiation 84 on nitrogen. Since 3H binds to a water molecule, it reaches the Earth's surface and the 85 subsurface through precipitation [27], [31], [32]. 3H also originates from anthropogenic 86 sources. Its activity concentrations in the environment increased by three orders of mag-87 nitude as a result of nuclear testing in the late 1950s. Leave just essential.

On figure 1 you should put groundwater flow direction so that you can connect with text in chapter 2. Materials and Methods when you are describing hydrogeological conditions

In table 1 add well depths

Since you did not measure with your instrument, somebody else measured for you, please delete the chapter 2.3 and just wrote in one sentence that Stable isotopes of 18O and 2H were measured in laboratory of Joanneum Research – AquaConSol GmbH, Graz, Austria and tritium activity concentration were measuredin the isotope laboratory HYDROSYS – Water and environmental protection developing Ltd, Budapest, Hungary.

Line 260 use Rozanski et al. Equitation for d-excess calculation

Table 2 missing delta annotation

Figure 3 is unclear due to very high oscillation of delta18O in precipitation; my suggestion is to divide ground and surface waters from precipitation so you can have four sub diagrams at one figure. It would be very nice to compare delta18O of groundwater with groundwater levels.

The same for tritium

This paragraph you should incorporate into Results and Discussion Oscillation of studied parameters at the sampled locations was observed together 214 with daily precipitation amounts in Cerklje/Brege and Puste Ložice. In the period before 215 the accumulation basin for the HPP Brežice was filled, 1100 mm of precipitation fell in 216 Pusti Ložice and 941 mm in Cerklje/Brege. At both precipitation stations, the share of 217 precipitation in spring, summer and autumn was similar (approximately 30%), while the 218 share of winter precipitation was only 10%. In the period after the accumulation basin for 219 the HPP Brežice was filled, 1174 mm of precipitation fell in Pusti Ložice and 995 mm in 220 Cerklje/Brege, which is 6% more than in the first period of isotope monitoring. At both 221 precipitation stations, the share of winter precipitation was the smallest (approximately 222 11%), the share of summer precipitation was the largest (approximately 45%), while a 223 similar amount of precipitation fell in spring and autumn (approximately 22%).

You did not make this so delete Geological data, hydrogeological data and the results of 207 groundwater flow simulations in the study site using a mathematical model were 208 obtained from Petauer et al. [1], [2], [3]. The model was created using VISUAL 209 MODFLOW 4.6.0.167 based on data from more than 50 boreholes in the study site, while 210 the simulation of groundwater flow was performed using the MODFLOW 2005 tool. 211 Based on the finite difference method, the model enabled the simulation of groundwater 212 flow in three directions at medium and high groundwater levels.

Rewrite Conclusion, give the essence of your research.

Reviewer 2 Report

The impact of the damming of the Sava river on aquifer recharge processes was studied using the evidence from hydrogen and oxygen isotopes. It’s a routine study and lack deep discussion. I recommend major revision, and the following comments should be taken into consideration by the authors. 

1.     The introduction part is poorly organized and should be structured better. The author should review and summarize the current research progress and scientific issues. Moreover, the novelty of the manuscript should be shown in the abstract and introduction part. Similarly, the significance of the Sava river should be strengthened.

2.     The decimal digits in tables and figures should be kept according to their analytical precision.

3.     The weakness of the manuscript is the discussion, which can be strengthened and more robust evidence should be given.

4.     Some of the Figures can be merged to a new figure.

5.     One of the aim was to integrate the isotopic results with the previous mathematical model results and discuss them in terms of different flow paths and possible sources of contamination. However, there is less discussion of this.

Round 2

Reviewer 1 Report

Dear Authors,

nice work. Only one minor remark, I did not observed first time, VSMOW is not producing for longer time it is VSMOW 2 and SLAP 2. 

Author Response

Dear reviewer,

thank you for valuable suggestions that helped us refine the manuscript. VSMOW2 and SLAP2 standards were added at the end of Subsection 2.2., line 309 and 310.